# Inductive Magnetic Nanoparticle Sensor Based on Microfluidic Chip Oil Detection Technology

**DOI:** 10.3390/mi11020183

**Published:** 2020-02-10

**Authors:** Chenzhao Bai, Hongpeng Zhang, Lin Zeng, Xupeng Zhao, Laihao Ma

**Affiliations:** College of Marine Engineering, Dalian Maritime University, Dalian 116026, China; baichenz@dlmu.edu.cn (C.B.); bob666zl@126.com (L.Z.); zhaoxp789@163.com (X.Z.); malaihao@dlmu.edu.cn (L.M.)

**Keywords:** inductance detection, magnetic nanoparticles, oil detection, microfluidic chip

## Abstract

The wear debris in hydraulic oil or lubricating oil has a wealth of equipment operating information, which is an important basis for large mechanical equipment detection and fault diagnosis. Based on traditional inductive oil detection technology, magnetic nanoparticles are exploited in this paper. A new inductive oil detection sensor is designed based on the characteristics of magnetic nanoparticles. The sensor improves detection sensitivity based on distinguishing between ferromagnetic and non-ferromagnetic wear debris. Magnetic nanoparticles increase the internal magnetic field strength of the solenoid coil and the stability of the internal magnetic field of the solenoid coil. During the experiment, the optimal position of the sensor microchannel was first determined, then the effect of the magnetic nanoparticles on the sensor’s detection was confirmed, and finally the concentration ratio of the mixture was determined. The experimental results show that the inductive oil detection sensor made of magnetic nanoparticle material had a higher detection effect, and the signal-to-noise ratio (SNR) of 20–70 μm ferromagnetic particles was increased by 20%–25%. The detection signal-to-noise ratio (SNR) of 80–130 μm non-ferromagnetic particles was increased by 16%–20%. The application of magnetic nanoparticles is a new method in the field of oil detection, which is of great significance for fault diagnosis and the life prediction of hydraulic systems.

## 1. Introduction

With the development of modern industrial technology, hydraulic systems and lubrication systems are widely used in mechanical automation, precision instruments, and other fields. However, in the normal operation of mechanical equipment, the system will experience wear due to the mutual friction of the surface and the invasion of external pollutants [1]. In order to improve economic efficiency, modern machinery usually works around the clock, so the wear and tear of the system will gradually accumulate during long-term production. If the wear and tear of the equipment is not discovered in time, it may have serious consequences.

Hydraulic oil is the “blood” of a hydraulic system. It can transfer energy, reduce relative friction, control the temperature of the system, and prevent the oxidation of the original work surface [2]. Therefore, we judge the wear and tear by obtaining information from the oil, and finally judge the state of the equipment. In hydraulic system failure, more than 75% of mechanical failures are caused by hydraulic oil failure [3]. The contaminants in hydraulic oil include internally generated contaminants and intruded contaminants—mainly solid particles, water, and air [4]. Contamination in hydraulic oil affects the working status of the entire hydraulic system. The normal allowable pollution size of the solid particles is below 20 μm. When abnormal mechanical wear occurs, the resulting solid particles rapidly increase in size, reaching even more than 100 μm. This can lead to dangerous working conditions and can cause huge losses [5,6,7].

Oil detection technology has been researched internationally. The particle counting method is a common method in the detection of solid particles in a fluid. It measures the signal pulse generated by the particle passing through the detecting device, and then determines the size distribution of the solid particles in the fluid according to the amplitude and quantity of the pulse. Therefore, people can truly achieve the accurate measurement of oil particle contamination.

Based on the different principles of particle-counting methods, a variety of pollutant particle detection methods have been developed. The acoustic detection method is based on the particle’s reflection amplitude of a sound wave. It can judge the particle count and size, but this method is easily affected by ambient temperature and noise, and it cannot distinguish the properties of solid particles [8,9,10,11,12]. An optical detection method was based on photoresistance or a light scattering counter to detect particles in oil. However, the detection accuracy of this method is affected by factors such as oil cleanliness, air bubbles in the oil, and so on, and it is impossible to distinguish between metal particles [13,14,15,16,17]. The resistance method is based on a particle meter principle proposed by WH Coulter in 1953 [18]. It detects current pulses generated when micropores pass through particles, but this method cannot distinguish between ferromagnetic and non-ferromagnetic metals, and the detection sensitivity low. The capacitance detection method consists of two electrodes that are close to each other to form a capacitor. When the particles pass between the two electrodes, the medium between the two electrodes is changed. This method cannot identify the properties of metal particles and is greatly affected by oil acid value and water [19,20,21]. The inductance detection method applies a high-frequency alternating current to the induction coil to generate a magnetic field in order to magnetize the metal particles. It can count and distinguish ferromagnetic and non-ferromagnetic metal particles by monitoring the change in the inductance value. This method is less affected by environmental factors, but the detection accuracy is lower [22,23,24,25]. Based on the inductive method, our team developed an inductive detection method based on microfluidic chips [26,27,28,29]. In our method, the inductive coil is in contact with the detection microchannel “0”, which greatly improves the detection precision of the inductance detection method.

Based on the previous inductive detection method formulated by our team, an inductive oil detection sensor with magnetic nanoparticles was designed. In contrast to the work of Liu et al. [30], they applied magnetics nanoparticles to the outside of the sensor, and we applied it to the inside of the sensing coil. The internal sensing unit was a combination of a solenoid coil and a magnetic nanoparticle layer. The magnetic nanoparticles designed in this paper were 10 nm Fe_3_O_4_, which is a black iron oxide nanoparticle, which has good biocompatibility, is superparamagnetic, and has a high magnetic field strength at room temperature [31]. We used the properties of the magnetic nanoparticles in combination with solenoid coils to improve the detection accuracy of inductive oil detection sensors.

## 2. Sensor Design and Fabrication

The design of the chip is shown in Figure 1. It is a whole-chip design, in which the microchannel had a diameter of 300 μm, and the particles were detected in the microchannel from the left to the right through the solenoid. As shown in Figure 1b, the sensor contained a coiled coil and a magnetic nanoparticle layer, the number of turns of the coil was 20 turns [32], and the diameter of the enameled wire was 60 μm. The magnetic nanoparticle layer was made of a mixture of magnetic nanoparticles and PDMS (polydimethylsiloxane), and the magnetic nanoparticle layer filled the inner region of the solenoid coil.

The micro-fabrication procedures are shown in Figure 2. When making a chip, the following steps were taken. First, the solenoid coil was wound by a precision winding machine (Shi Li SRDZ23-1B, Zhong Shan Shi Li Wire Winder Equipment, Zhong Shan, China), and multiple sets were wound, and the winding direction and parameters of each coil were the same. Then PDMS, and coagulant were configured according to a 10:1 ratio, and a small amount of configured PDMS was taken out and uniformly mixed with magnetic nanoparticles. PDMS and magnetic nanoparticles were mixed at a ratio of 1:2. Casting mixed materials were then cast into spiral coils and heat-cured at 80 °C to form a magnetic nanoparticle layer, and the magnetic nanoparticles layer was perforated to reserve microchannel holes. Next, a 7 cm long, 300 μm diameter microchannel-forming mold was selected, which was passed through the microchannel holes of the coil and attached to the slide. Finally, the slide was cast with PDMS and placed in an 80 °C oven for heat curing. We extracted the microchannel mold to form a 300-μm microchannel, punched the oil inlet and the oil outlet, and the sensor was fabricated.

## 3. Analysis of Detection Principle

The solid particles were magnetized in the magnetic field inside coil D_1_ in Figure 1, and the magnetized particles are shown in Figure 3. It is divided into an external magnetization field and an internal magnetization field. The external magnetization field is equivalent to a magnetic dipole whose magnetic dipole moment is:(1)$$P=u_{0}m=u_{0}MV$$
where M is the magnetization, m is the overall magnetic moment of the particle, V is the volume of the particle, and $u_{0}$ is the magnetic permeability. At this stage, the internal magnetic field of the particle is:(2)$$H_{in}=-NM\;(0<N<1)$$
(3)$$B_{in}=u_{0}H_{in}+u_{0}M$$
where $H_{in}$ is the internal demagnetizing field of the particle, $B_{in}$ is the internal magnetic induction intensity of the particle, and N is the demagnetization factors. If the electric field is stable, the internal magnetic field strength of the coil is $H_{0}$, then the combined magnetic field of the particle in the microchannel is:(4)$$H=H_{0}+H_{in}=H_{0}-NM$$

Oil as a linear medium:(5)$$M=\chi_{m}H$$
(6)$$\chi_{m}=u_{r}-1$$

$\chi_{m}$ is the susceptibility of the oil, $u_{r}$ is the relative permeability. Then there is:(7)$$M=\frac{u_{r}-1}{1+N\left(u_{r}-1\right)}H_{0}$$

According to the above formula, the degree of magnetization of the particles is related to $H_{0}$, and $H_{0}$ is the internal magnetic field of coil D_1_ in Figure 1. Since the ferromagnetic particles have a relatively high magnetic permeability, when they are magnetized by the magnetic field region the particles were magnetized to generate the same magnetic field as the original magnetic field. $H_{in}$ is the same as the original magnetic field $H_{0}$, so the inductance value is increased. The relative magnetic permeability of the non-ferromagnetic particles is low, and the particles generate a strong electric eddy current while being magnetized, and the direction of the eddy current is opposite to the direction of the coil magnetic field. $H_{in}$ is opposite to the original magnetic field $H_{0}$, so the inductance value is reduced.

The magnetization was directly affected by the sensor’s ability to detect particles. This property was used. We combined magnetic nanoparticles and solenoid coils to improve the detection sensitivity of inductive oil detection sensors. Magnetic nanoparticles filled the inside of the coil to form a magnetic nanoparticle layer. The length of the layer is D_1_ and the diameter is D_2_. When manufacturing, the length and diameter of the layer need to be consistent with the coil. The microchannel in the center of the coil was made first, and the optimal microchannel position will be discussed in the experimental section. The magnetic nanoparticles and PDMS should be completely mixed to ensure the uniformity of the layer. Magnetic nanoparticles can exhibit strong magnetic induction in the presence of external magnetic fields. The mechanism of action of the magnetic nanoparticles is shown in Figure 3.

In Figure 3, the magnetic field strength of the magnetic nanoparticles is $H_{1}$, and the total magnetic field strength of the spatial magnetic field is $H^{\prime }$:(8)$$H^{\prime }=H_{0}+H_{1}$$

For iron particles, the relative permeability is $u_{r}\gg 1$. Therefore, the magnetization is greater than the eddy current. At this time, the iron particles are dominated by the magnetization in the alternating electromagnetic field. The magnetization of the iron particles is enhanced to produce the same effect as the original magnetization direction, so that the detection signal value is increased upward. So, according to Equation (7), the magnetization M is: (9)$$M=\frac{u_{r}-1}{1+N\left(u_{r}-1\right)}H\prime$$

Due to the action of the magnetic nanoparticles, the total magnetic field is increased, so the magnetization M is increased and the magnetization of the particles is enhanced.

For copper particles, the relative permeability is $u_{r}=1$. Therefore, the eddy current is greater than the magnetization. In this occasion, the copper particles are dominated by the eddy current in the alternating electromagnetic field. The eddy current will resist the original magnetic field and produce an effect opposite to the direction of the magnetic field, and the detection signal value is decreased. So according to Equation (8), when the total magnetic field $H^{\prime }$ is increased in the coil, according to Faraday’s law of electromagnetic induction, it can be said that the larger the magnetic field, the larger the induced electromotive force, and the stronger the eddy current effect. The eddy current effect is strengthened, and the copper particles will produce a larger signal. The larger the size of the copper particles, the stronger the effect of the eddy current. Thus, the detection capability of the sensor is improved.

## 4. Experiment and Data Analysis

The instruments used in the experiment included an impedance analyzer (Keysight E4980A, Agilent Technologies Inc., Bayan Lepas, Malaysia), a microscope (Nikon AZ100, Nikon, Tokyo, Japan), a micro-injection pump (Harvard Apparatus B-85259, Harvard Apparatus, Holliston, MA, USA), a computer with a LabVIEW data acquisition unit, and our sensor. The experimental instruments were connected before the experiment. The test bench system is shown in Figure 4.

### 4.1. Selection of Microchannel Position

Before the experiment, the impedance analyzer was preheated for 30 min, and the voltage was set to 2 V and the frequency to 2 MHz. The flow rate of the microinjection pump was adjusted to 30 μL/min. It has been proven that the smaller the flow rate of the oil, the larger the detection signal value, so we chose a minimum flow rate of 30 μL/min [33]. The sieved iron particles were represented by ferromagnetic particles. The sizes of the iron particles were 20, 30, 40, 50, 60, and 70 μm. The particles of each size were weighed to 5 mg and mixed with 120 mL of hydraulic oil. Like the iron particles, non-ferromagnetic particles were represented by copper particles. Copper particles of 80, 90, 100, 110, 120, and 130 μm were prepared, weighed as 6 mg separately, and mixed with 120 mL of hydraulic oil. The mixed oil was placed on an ultrasonic oscillator (IKA S25, IKA, Staufen, Germany) and shaken for 2 min. Samples of different sizes of mixed oil were taken out and placed in a microinjection pump for use.

We first performed the determination experiment for the microchannel position. The internal space magnetic field changed due to the addition of the magnetic nanoparticle layer inside the solenoid coil. When the microchannel was in different positions, the detection had different experimental effects. The prepared sensor with a magnetic nanoparticle layer was applied to the experiment. As shown in Figure 5, the microchannel had three positions for experimental verification, namely, positions 1, 2, and 3. The three positions of the microchannel were compared. Different sizes of iron particles and copper particles were detected. The test results are shown in Figure 6 and Figure 7.

The signal-to-noise ratio (SNR) expresses the detection capability of the sensor. The larger the value of the SNR, the stronger the detection effect, and vice versa. The formula for calculating the SNR is:(10)$$SNR=\frac{Signal\;value}{Noise\;value}$$

The signal value is the maximum value of the signal minus the average noise value. The noise value equals the average maximum minus the average minimum, as shown in Figure 8.

Based on the results shown in Figure 6 and Figure 7, the experimental results were obtained according to the different positions shown in Figure 5. When the microchannel was in position 1, its detection SNR was the highest. When the microchannel was in position 3, its detection SNR was the lowest. The reason is that the magnetic field generated by the coil was relatively weak. The magnetic field closer to the edge of the coil was stronger, and the magnetic field closer to the central axis was weaker. Although we added magnetic nanoparticles inside the coil, the magnetic field distribution inside the coil did not change. In Figure 6 and Figure 7, iron particles and copper particles were best detected at position 1, so it can be determined from this experiment that position 1 was the best experimental position. Therefore, follow-on experiments were carried out on the basis of position 1.

### 4.2. Sensors Detection Effect Comparison

Position 1 was verified as the best position for a microchannel in a sensor. Next, a sensor with or without a magnetic nanoparticle layer was selected for comparison experiments, and the inside of the two sensor coils is shown in Figure 9. Both sensors were fabricated using the same process and the experimental parameters were consistent. Figure 10 shows the two signal values of 50 μm iron particles obtained under different sensors. Figure 11 shows the two signal values of 110 μm copper particles obtained under different sensors.

As can be seen from the above figures, in which the blue lines show the detection signal values of the magnetic nanoparticles, the signal values of the ferromagnetic and non-ferromagnetic particles were significantly improved compared with the sensor without the magnetic nanoparticle layer—the detection effect of the nanoparticle layer was better. Figure 12 shows the detected signal-to-noise ratios (SNRs) of 20–70 μm iron particles. Figure 13 shows the detected signal-to-noise ratios (SNRs) of 80–130 μm copper particles.

In Figure 12 and Figure 13, the sensor with magnetic nanoparticles had a higher signal-to-noise ratio (SNR) for all particle sizes than the sensor without magnetic nanoparticles. This experiment proves that magnetic nanoparticles have a beneficial effect on sensor detection.

Table 1 shows the results, after many experiments and calculations. For example, for the 50 μm iron particles, the sensor without magnetic nanoparticles had a base inductance of 1.5135 × 10^−6^ H and had a signal value of 3.1167 × 10^−10^ H. Meanwhile, the sensor with magnetic nanoparticles had a base inductance of 1.4539 × 10^−6^ H and the signal value of 3.6122 × 10^−10^ H. This shows that the magnetic nanoparticles not only reduced the basic inductance value, but also increased the signal value, causing the signal-to-noise ratio (SNR) to increase.

The detection accuracy of the sensor with the magnetic nanoparticle layer improved by 20%–25% when detecting 20–70 μm iron particles. When detecting 80–130 μm copper particles, the detection accuracy was improved by 16%–20%.

### 4.3. Concentration Comparison of Magnetic Nanoparticles

The nanoparticles and PDMS were mixed according to the volume ratio. When the concentration of magnetic nanoparticles mixed with PDMS was greater than 2:1, the resulting mixture did not form a solid, so we chose the maximum mix ratio of 2:1. Figure 14 shows the mixed concentrations of magnetic nanoparticles and PDMS, and their SEM images. According to the SEM results, the internal morphology of the mixtures with different concentrations after forming a solid was different. The higher the concentration, the more uniform the distribution of magnetic nanoparticles. The experimental verification was as follows.

In the experiment, 50-μm iron particles and 110-μm copper particles were selected for testing. Five sets of data were measured for each particle, and then the average signal value, signal-to-noise ratio, and error value were calculated.

As shown in Figure 15 and Figure 16, the mixing ratio of the magnetic nanoparticles to the PDMS in group (a) was 2:1, and the detection signal value was the largest, the signal-to-noise ratio was the highest, and the error value was relatively low. The concentration ratios of group (b) and group (c) were 1:1 and 0.5:1, respectively, but their detection signal values and signal-to-noise ratios were not very different, and the detection sensitivity was not improved. Therefore, according to the experimental results, when the concentration ratio was 2:1, the detection accuracy was the highest and the effect was the best. This echoes the SEM results.

## 5. Results and Discussion

Inductive oil detection sensors with magnetic nanoparticles were designed and fabricated based on the induction of a single solenoid sensor. The characteristics of magnetic nanoparticles were applied, which improved the detection of the sensor. The experiment first verified the optimal position of the microchannel and then compared the detection capabilities of the two sensors. Finally, the mixed concentration of magnetic nanoparticles and PDMS was determined. The experimental results show that the microchannels had the best detection effect when they were close to the inner wall of the coil, and, with the magnetic nanoparticle layer sensor, could increase the signal-to-noise ratio (SNR) by 20%–25% for ferromagnetic particles and by 16%–20% for non-ferromagnetic particles.

In this article, magnetic nanoparticles were innovatively applied to the field of oil detection sensors. This was a brand new but imperfect attempt. There are some issues worth discussing. Compared with other related works, the detection accuracy of the inductive sensor designed by the University of Akron [15] was 55 μm iron particles. The sensor designed by BUAA (Beihang University) [34] had a detection accuracy of 81 μm iron particles, and the study did not involve non-ferromagnetic particles. The inductive sensor designed by USTC (University of Science and Technology of China) could detect 130 μm iron particles and 230 μm copper particles [35]. The sensor designed by us could detect 20 μm iron particles and 80 μm copper particles. The sensor we designed has the advantages of a small size and a high detection accuracy. However, future work is required to improve the detection throughput. This work has a great significance for fault diagnosis and the online life prediction of hydraulic systems.

## Figures and Tables

**Figure 1 micromachines-11-00183-f001:**
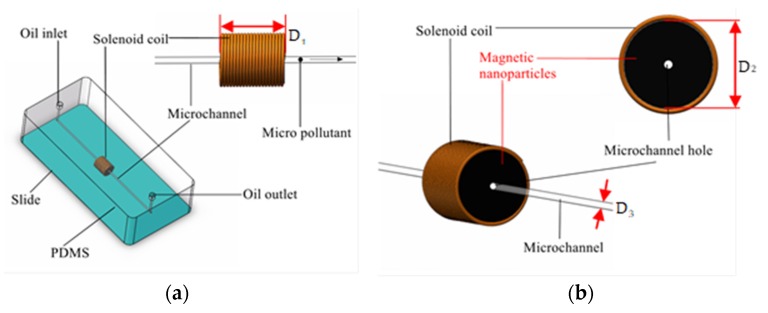
(**a**) Design of the sensor. The length of the solenoid coil D_1_ = 2 mm. (**b**) Sensing unit. Coil inner diameter D_2_ = 1.2 mm, microchannel diameter D_3_ = 300 μm.

**Figure 2 micromachines-11-00183-f002:**
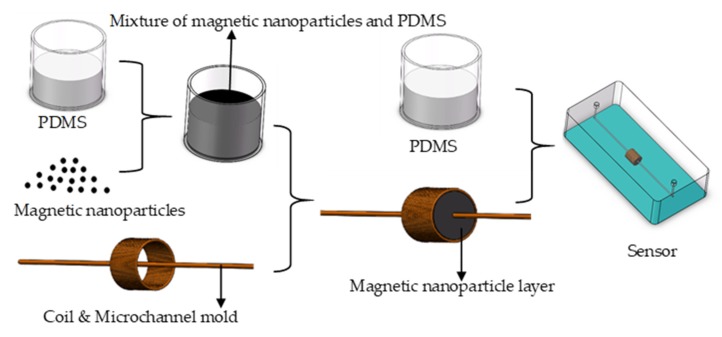
The fabrication of the sensor with magnetic nanoparticles.

**Figure 3 micromachines-11-00183-f003:**
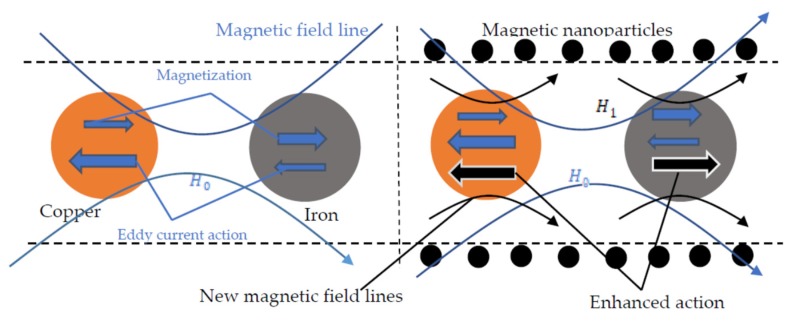
Magnetization effect with magnetic nanoparticles.

**Figure 4 micromachines-11-00183-f004:**
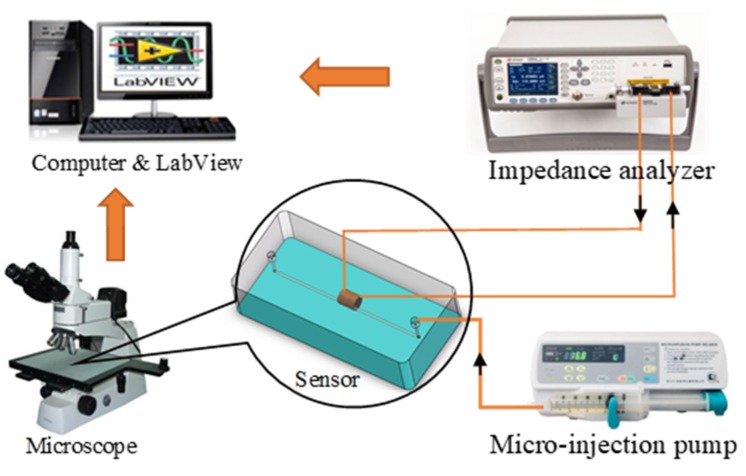
The experiment system.

**Figure 5 micromachines-11-00183-f005:**
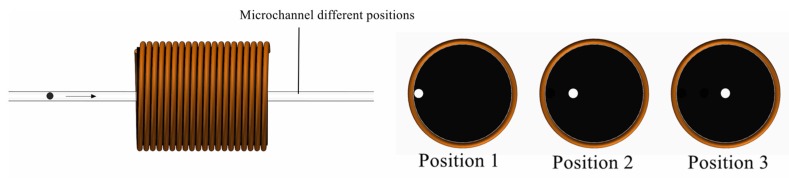
Different positions of microchannels.

**Figure 6 micromachines-11-00183-f006:**
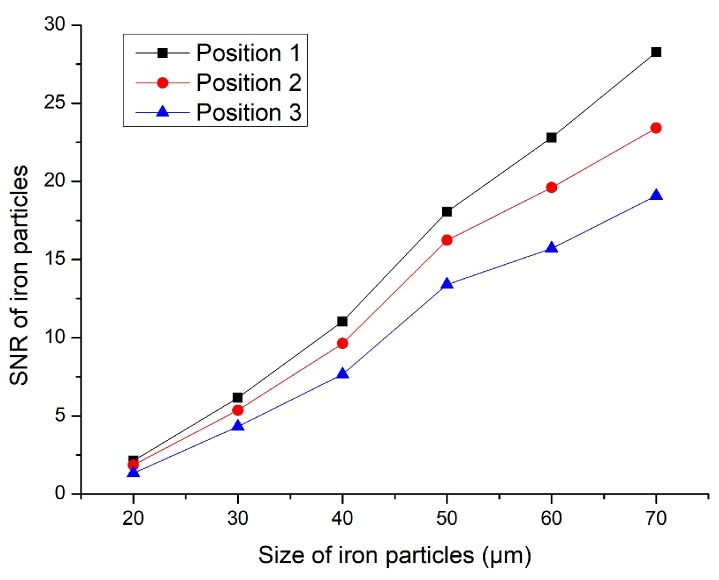
Signal-to-noise ratio (SNR) of iron particles.

**Figure 7 micromachines-11-00183-f007:**
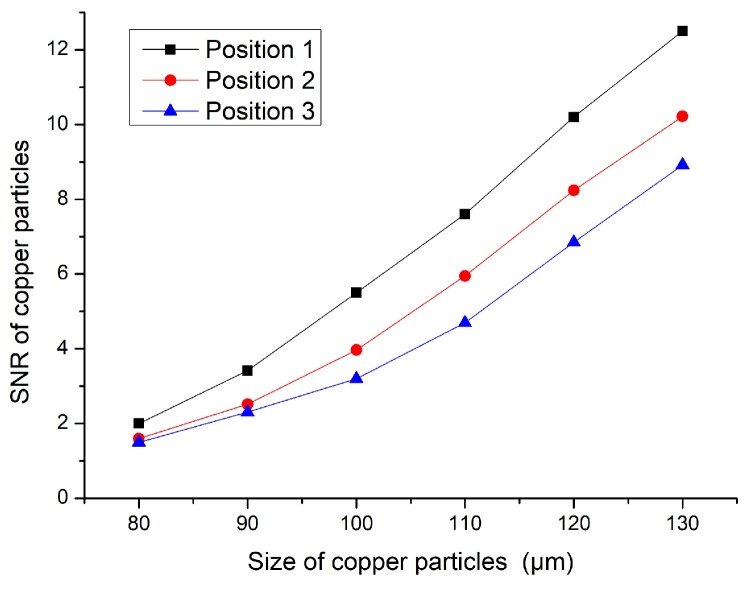
SNR of copper particles.

**Figure 8 micromachines-11-00183-f008:**
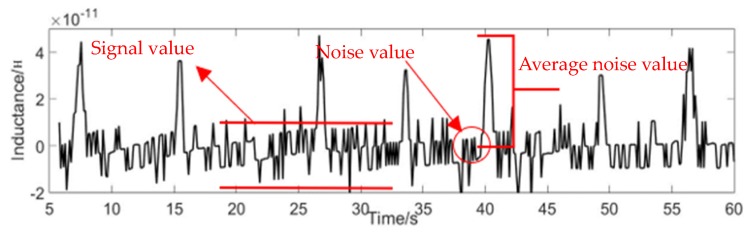
Calculation of SNR.

**Figure 9 micromachines-11-00183-f009:**
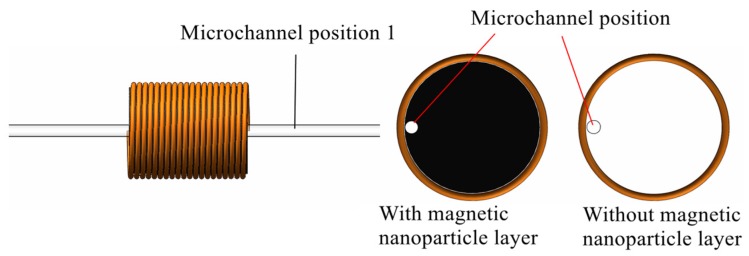
With or without magnetic nanoparticles.

**Figure 10 micromachines-11-00183-f010:**
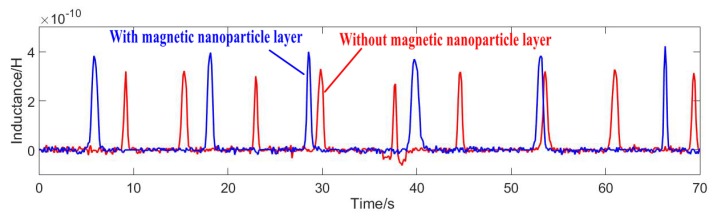
Signal comparison: 50μm iron particles.

**Figure 11 micromachines-11-00183-f011:**
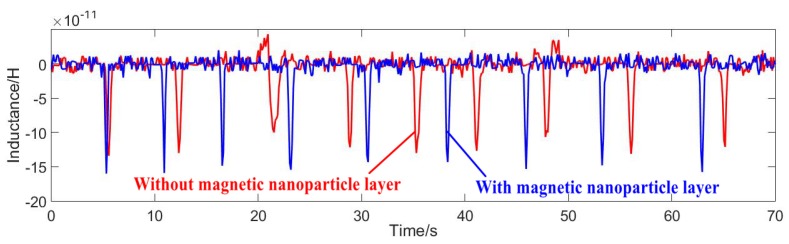
Signal comparison: 110 μm copper particles.

**Figure 12 micromachines-11-00183-f012:**
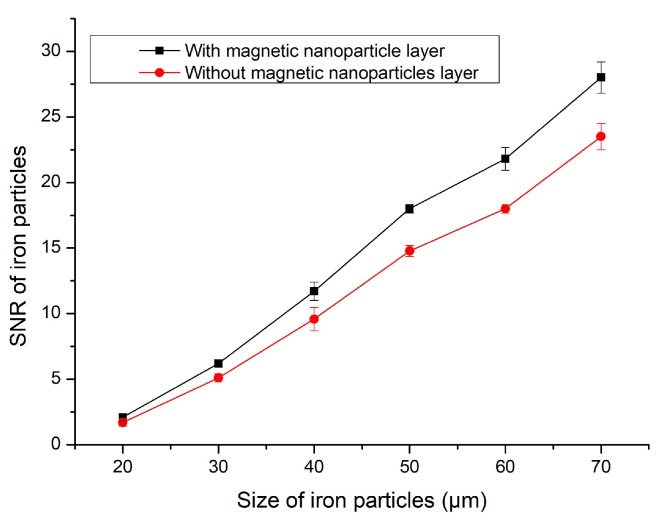
SNR of 20–70 μm iron particles.

**Figure 13 micromachines-11-00183-f013:**
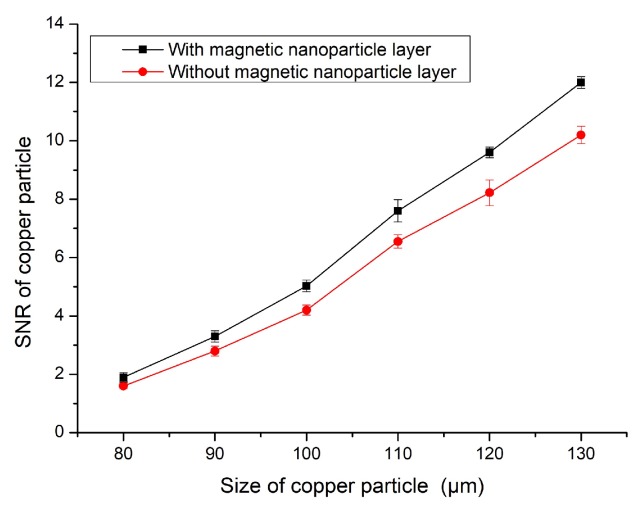
SNR of 80–130 μm copper particles.

**Figure 14 micromachines-11-00183-f014:**
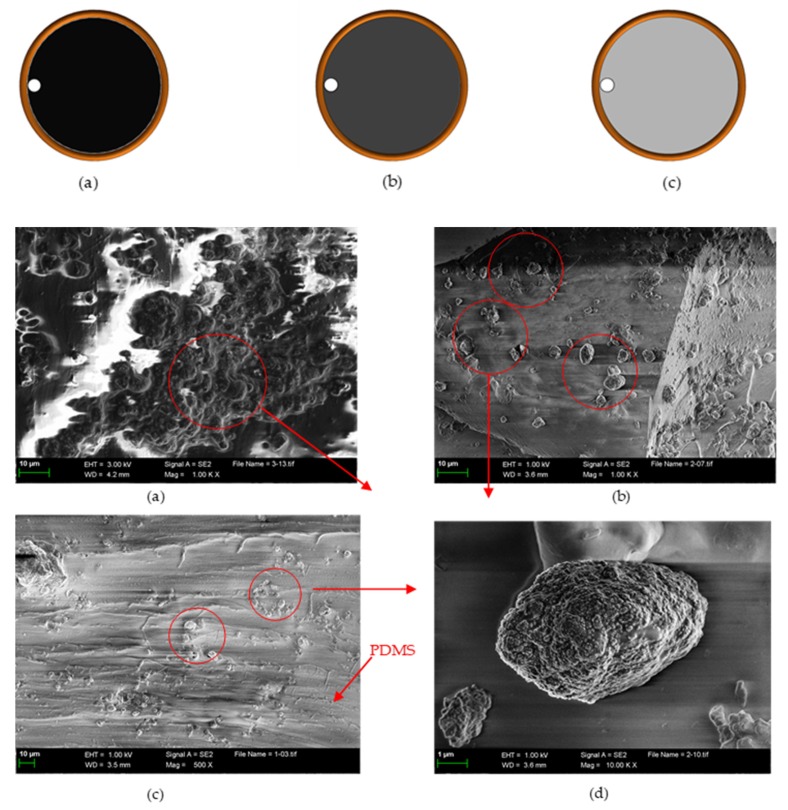
Magnetic nanoparticle concentration comparison and SEM. (**a**) Polydimethylsiloxane (PDMS) and magnetic nanoparticles mixed at a ratio of 1:2, large content of magnetic nanoparticles in SEM, uniform distribution; (**b**) PDMS and magnetic nanoparticles mixed at a ratio of 1:1, sparse and uneven distribution in SEM; (**c**) PDMS and magnetic nanoparticles mixed at a ratio of 2:1, almost no distribution in SEM; (**d**) A magnetic nanoparticle in PDMS, composed of many magnetic nanoparticle aggregates.

**Figure 15 micromachines-11-00183-f015:**
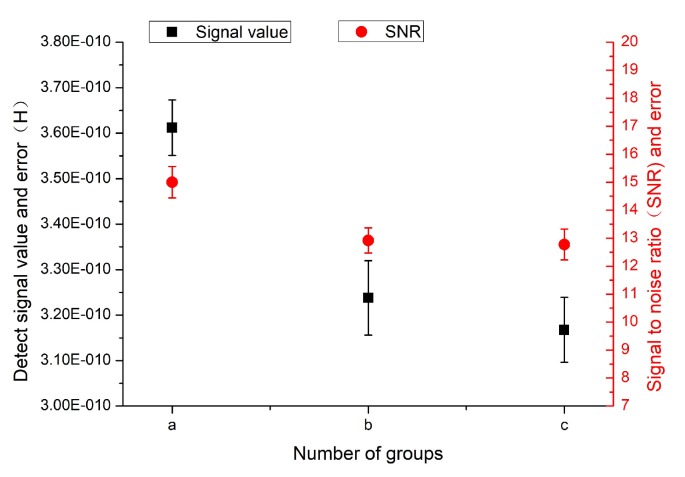
Detection results for 50-μm iron particles. “a”, “b”, and “c“ correspond to the treatments labeled with the same letters in Figure 14.

**Figure 16 micromachines-11-00183-f016:**
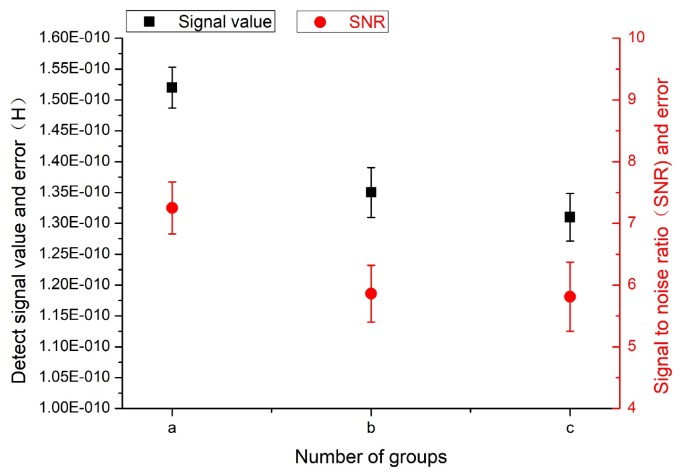
Detection results for 110-μm copper particles. “a”, “b”, and “c“ correspond to the treatments labeled with the same letters in Figure 14.

**Table 1 micromachines-11-00183-t001:** Comparison of two sensors for detecting particles.

Type of Particle (μm)	Sensor without Magnetic Nanoparticles		Sensor with Magnetic Nanoparticles		SNR Increase
	Basic Inductance Value (H)	Signal Value (H)	Basic Inductance Value (H)	Signal Value (H)	
20 μm iron	1.5029 × 10^−6^	4.1075 × 10^−11^	1.4109 × 10^−6^	4.4155 × 10^−11^	24%
50 μm iron	1.5135 × 10^−6^	3.1167 × 10^−10^	1.4539 × 10^−6^	3.6122 × 10^−10^	22%
80 μm copper	1.4723 × 10^−6^	3.2502 × 10^−11^	1.4909 × 10^−6^	4.4115 × 10^−11^	19%
110 μm copper	1.4917 × 10^−6^	1.3067 × 10^−10^	1.5052 × 10^−6^	1.5142 × 10^−10^	20%

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
