# Peer review of "Inductive Magnetic Nanoparticle Sensor Based on Microfluidic Chip Oil Detection Technology"

_micromachines, 2020, doi:10.3390/mi11020183_

Round 1
Reviewer 1 Report
This paper is much improved from its last version. I suggest to publish this paper after English Proof Reading.
Author Response
Dear reviewer. Thank you very much for your review.
Reviewer 2 Report
The authors demonstrate that improvements to the signal-to-noise ratio (SNR) of their microfluidic-chip-based inductive sensor for wear debris in hydraulic oil can be made by immobilizing Fe3O4 nanoparticles within the solenoid. The enhancements in SNR extend to 25% for 20-70 μm ferromagnetic (iron) particles, and to 20% for 80-130 μm non-ferromagnetic (copper) particles. The utilization of magnetic particles to increase the magnetic field produced by a coil has been published previously, but this may be the first description of magnetic nanoparticles employed in an inductive sensor for hydraulic oil. The authors employed appropriate controls in the experiments, and performed optimization of the sensor configuration and nanoparticle concentration. The sensor geometry, microfluidic chip preparation, and key results are clearly presented in the figures.
With a few revisions, I recommend that the manuscript be published in Micromachines:
Major Revisions
A more complete description of the inductive method for detection of particles in oil should be given in the introduction. Previous work on utilization of magnetic particles to enhance the magnetic field within a solenoid, such as that recently reported by Liu et al. (Micromachines 2019, 10, 440), should be acknowledged. The geometrical and process factors determining the measured induction should be explained in the theory section.
Minor Revisions
The unit is incorrect in the vertical axis of Figure 8. Two figures are numbered as 11. Appropriate units should be supplied for Table 1 and its discussion.
Author Response
Dear Reviewer
Thank you very much for your valuable comments and suggestions, these are very helpful to improve the paper. We responded to each of your comments, and corrected some errors and added some content in the paper. In order to distinguish from the previous changes, all new changes are in blue font. We take your suggestions seriously and take the article to a new level. Article grammar and structure revised by MDPI editing service. If you do not satisfy with our response and revision, please do not hesitate to contact us. We will seriously modify it again. Thank you again.
Reviewer Major Revisions: A more complete description of the inductive method for detection of particles in oil should be given in the introduction. Previous work on utilization of magnetic particles to enhance the magnetic field within a solenoid, such as that recently reported by Liu et al. (Micromachines 2019, 10, 440), should be acknowledged. The geometrical and process factors determining the measured induction should be explained in the theory section.
Response: Thank you very much for your comments. We have modified the introduction and added resistance detection in the oil detection particle counting method. The work of Liu et al. is cited and compared with ours. Geometric and technological factors are explained in the theoretical section.
Reviewer Minor Revisions: The unit is incorrect in the vertical axis of Figure 8. Two figures are numbered as 11. Appropriate units should be supplied for Table 1 and its discussion
Response: Thank you very much for your comments. We have modified the relevant content.
Reviewer 3 Report
Before the publication on Micromachines:
1) Authors have to improve the abstract where it is not clear what the sensor has to detect. Oil detection is not exhaustive.
2) Figure 2 has to be improved because is not clear.
3) Figure 4 is not clear and a drawing of the measurement setup is needed
4) In order to enhance the motivations, more details on possible applications are needed
5) the SNR is already high for the considered particles, why to improve this parameter? It would be more significant to improve the SNR when it was very low. Why the author have not considered particles with low SNR?
Author Response
Dear Reviewer
Thank you very much for your valuable comments and suggestions, these are very helpful to improve the paper. We responded to each of your comments, and corrected some errors and added some content in the paper. In order to distinguish from the previous changes, all new changes are in blue font. We take your suggestions seriously and take the article to a new level. Article grammar and structure revised by MDPI editing service. If you do not satisfy with our response and revision, please do not hesitate to contact us. We will seriously modify it again. Thank you again.
1) Authors have to improve the abstract where it is not clear what the sensor has to detect. Oil detection is not exhaustive.
Response: Thank you very much for your comments. The abstract has been modified with some application details.
2) Figure 2 has to be improved because is not clear.
Response: Thank you very much for your comments. Related content has been modified.
3) Figure 4 is not clear and a drawing of the measurement setup is needed
Response: Thank you very much for your comments. New detection system diagram is drawn.
4) In order to enhance the motivations, more details on possible applications are needed
Response: Thank you very much for your comments. An increase in the flow velocity of the particles results in a decrease in the detection signal value. In the future, we can solve the problem of motivations by increasing the throughput. The sensor can use a multi-channel method, increase the diameter of the channel, or design a circular channel to increase the throughput of the oil, but this will cause a problem of reduced detection accuracy. Increasing the oil throughput is necessary, and it is the work we will explore.
5) the SNR is already high for the considered particles, why to improve this parameter? It would be more significant to improve the SNR when it was very low. Why the author have not considered particles with low SNR?
Response: Thank you very much for your comments. In this paper, a magnetic nanoparticles sensor is designed by combining magnetic nanoparticles and inductor coils. The goal is to increase the magnetic field in the detection area and detect smaller particles. The particles with high SNR are used because the particle size is relatively large and regular, and the error is relatively low. Selecting particles with high SNR in experiments at the best detection location is also to reduce experimental errors. It is found through experiments that magnetic nanoparticles have a relatively obvious effect of increasing the magnetic field. Then we used a new sensor to determine the lower limit of detection. 20 μm iron particles and 80 μm copper particles, which are particles with low SNR. We did relevant comparison experiments (Table 1). The SNR of 20 μm iron particles increased by 24%, and the SNR of 80 μm copper particles increased by 19%.
Round 2
Reviewer 2 Report
One minor point: On line 74, the magnetic particles used by Liu et al. (Micromachines 2019, 10, 440) are microparticles (800 mesh), not nanoparticles.
Reviewer 3 Report
I recommend the publication on Micromachines